Toxicity of clothianidin to common Eastern North American fireflies

Pearsons Kirsten Ann kfp5094@psu.edu 1
Lower Sarah E. 2
Tooker John F. 1
1 Department of Entomology, Pennsylvania State University , University Park , PA , United States of America
2 Biology Department, Bucknell University , Lewisburg , PA , United States of America
Anderson Todd
Electronic publication date: 2021 Nov 19
Publication date: 2021
Volume: 9
Electronic Location ID: e12495
Received 2021 Sep 16; Accepted 2021 Oct 25
Copyright: ©2021 Pearsons et al.
Copyright year: 2021
Copyright holder: Pearsons et al.
License: This is an open access article distributed under the terms of the Creative Commons Attribution License, which permits unrestricted use, distribution, reproduction and adaptation in any medium and for any purpose provided that it is properly attributed. For attribution, the original author(s), title, publication source (PeerJ) and either DOI or URL of the article must be cited.
License URL: https://creativecommons.org/licenses/by/4.0/

Keywords: Neonicotinoid, Clothianidin, Flirefly, Lampyridae, Acute toxicity, Long-term toxicity, Photuris versicolor, Photinus pyralis

Funding: National Institute of Food and Agriculture and Hatch Appropriations #PEN04606 #1009362 This work was funded by the College of Agricultural Sciences at Penn State via the National Institute of Food and Agriculture and Hatch Appropriations under Project #PEN04606 and Accession #1009362. The funders had no role in study design, data collection and analysis, decision to publish, or preparation of the manuscript.

==============================
Background

Previous research suggests that fireflies (Coleoptera: Lampyridae) are susceptible to commonly used insecticides. In the United States, there has been a rapid and widespread adoption of neonicotinoid insecticides, predominantly used as seed coatings on large-acreage crops like corn, soy, and cotton. Neonicotinoid insecticides are persistent in soil yet mobile in water, so they have potential to contaminate firefly habitats both in and adjacent to application sites. As a result, fireflies may be at high risk of exposure to neonicotinoids, possibly jeopardizing this already at-risk group of charismatic insects.

Methods

To assess the sensitivity of fireflies to neonicotinoids, we exposed larvae of Photuris versicolor complex and Photinus pyralis to multiple levels of clothianidin-treated soil and monitored feeding behavior, protective soil chamber formation, intoxication, and mortality.

Results

Pt. versicolor and Pn. pyralis larvae exhibited long-term intoxication and mortality at concentrations above 1,000 ng g−1 soil (1 ppm). Under sub-lethal clothianidin exposure, firefly larvae fed less and spent less time in protective soil chambers, two behavioral changes that could decrease larval survival in the wild.

Discussion

Both firefly species demonstrated sub-lethal responses in the lab to clothianidin exposure at field-realistic concentrations, although Pt. versicolor and Pn. pyralis appeared to tolerate higher clothianidin exposure relative to other soil invertebrates and beetle species. While these two firefly species, which are relatively widespread in North America, appear somewhat tolerant of neonicotinoid exposure in a laboratory setting, further work is needed to extend this conclusion to wild populations, especially in rare or declining taxa.

Introduction

In the United States alone, insects are estimated to provide over $50 billion in ecological services (Losey & Vaughan, 2006). Fireflies have great popular appeal and aesthetic and cultural value, but fireflies also contribute biological control of some pest species, including slugs and snails, which can be important agricultural pests (Godan, 1983; Lewis, 2016). Human activities, however, have put these services at risk by triggering global insect declines (Sánchez-Bayo & Wyckhuys, 2019). Some charismatic groups such as fireflies (Coleoptera: Lampyridae) may be at elevated risk of at least localized extinction due to ongoing human activities such as heavy pesticide use in and around their habitats (Reed et al., 2020).

Despite broad agreement that pesticides can pose a serious extinction threat to fireflies (Lewis et al., 2020), there is a very poor understanding of the direct toxicity of insecticides on fireflies. The most commonly applied classes of insecticides (neonicotinoids, pyrethroids, and organophosphates) are broadly neurotoxic to most insect taxa (Sparks, 2013), so fireflies are unlikely to be an exception. Indeed, full-strength organophosphate and neonicotinoid formulations are toxic to the aquatic firefly larvae Luciola cruciata and Luciola lateralis, respectively (Tabaru et al., 1970; Lee et al., 2008). Unfortunately, there have been no studies assessing how terrestrial firefly larvae respond to residual concentrations of these insecticides in soil, a likely route of exposure. Larvae of many common firefly species in North America are soil-dwellers that intimately interact with soil as they forage for prey and form protective molting chambers (Buschman, 1984; Lewis, 2016). These larvae inhabit forested, suburban, and agricultural soils, where neonicotinoid insecticides are often applied directly, or via coatings on crop seeds, to protect against pests (Knoepp et al., 2012; Douglas & Tooker, 2015; Simon-Delso et al., 2015). In these habitats, neonicotinoid concentrations in soil can range from less than 5 ng g−1 to over 4,000 ng g−1 (Knoepp et al., 2012; Schaafsma et al., 2015; Pearsons et al., 2021), concentrations that could plausibly influence behavior and survival of firefly larvae (Lee et al., 2008). Some indirect evidence suggests that firefly larvae are susceptible to neonicotinoids because adult lampyrid densities have been found to be lower where neonicotinoid-coated seeds were planted (Disque et al., 2019); however, to our knowledge, there have been no direct evaluations of how terrestrial firefly larvae respond to neonicotinoid-treated soil.

To assess the direct sensitivity of fireflies to neonicotinoid insecticides, we measured feeding behavior, development, and survival of larvae of two common North American firefly species—Photuris versicolor species complex and Photinus pyralis (Linnaeus 1767)—exposed to clothianidin-treated soil. We focused on clothianidin, one of the most widely-used seed- and soil-applied neonicotinoid and the primary metabolite of another commonly applied neonicotinoid, thiamethoxam (Douglas & Tooker, 2015). Generally applied to combat sucking and chewing insects, clothianidin disrupts insect central nervous systems, leading to paralysis and death (Simon-Delso et al., 2015). We exposed larvae to multiple field-realistic levels of clothianidin-treated soil for 30 to 100 days with the expectation that they would be sensitive to clothianidin at concentrations that have been detected in firefly habitats.

Materials & Methods

Chemicals

We acquired clothianidin from Chem Service (West Chester, PA, USA; purity ≥ 98%), and prepared stock solutions of 2  × 102, 2  × 103, 2  × 104, and 2  × 105 ng mL−1 clothianidin in acetone (Sigma Aldrich, St. Louis, MO, USA, ACS reagent, purity ≥ 99.5%). Pure acetone served as a control. We stored stock solutions at 4 °C and allowed them to reach room temperature (20 °C) before applying them to soils for the assays.

Firefly collection and colony care

We ran toxicity assays on three separate cohorts of fireflies (Table 1): late-instar larvae from the Photuris versicolor species complex (Pt. Photuris), early-instar Pt. versicolor, and early-instar Photinus pyralis (Pn. pyralis). Both Pt. versicolor and Pn. pyralis are relatively large-bodied (6–20 mm adult body length), widespread firefly species found throughout Eastern North America, and their populations do not appear to be declining (Lewis, 2016). Because both species spend 1–2 years in the soil as larvae and feed on soil invertebrates (Pt. versicolor are thought to feed on a diversity of soil invertebrates whereas Pn. pyralis larvae are considered specialists on earthworms; (McLean, Buck & Hanson, 1972; Buschman, 1984; Lewis, 2016)), they likely experience chronic contact and oral neonicotinoid exposure in contaminated habitats.

Table 1 Summary details of the firefly larvae (three cohorts of two species) used for toxicological assays.

Species	Instar
group	Age
(months)	Mean weight ±  SD
(mg)	# used
in assay	Source	
Pt. versicolor	late	>12	77 ± 17	30	wild-collected	
	early	<3	9 ± 3	15	lab-reared	
Pn. pyralis	early	<3	1.4 ± 0.5	75	lab-reared	

Five of the late-instar Pt. versicolor were reared from eggs laid by a mated female collected in late July 2019 from the Bucknell University Chillisquaque Creek Natural Area (Montour Co, PA; 41°01′15″N, 76°44′53″W) , while the other 25 late-instar Pt. versicolor were wild-collected in summer of 2019 from multiple locations throughout Pennsylvania: Bald Eagle State Park (5 August; Centre Co, 41°00′44.0″N 77°12′54.3″W), Allegheny National Forest (24–25 June; Forest Co, 41°31′29.8″N 79°17′33.9″W), and Bucknell University Forrest D. Brown Conference Center (23–24 July; Union Co, PA; 40°57′28″N, 77°00′49″W). Specimens were collected in Bald Eagle State Park, Pennsylvania under permit SFRA-1907 to S. Lower and in Allegheny National Forest under firefly monitoring permit to the Pennsylvania Firefly Festival. Larvae were wild-collected at night by visually inspecting the ground for their faint glows. Larvae were identified to genus by external morphology (McLean, Buck & Hanson, 1972). After collection, we housed individual larvae in 16-oz clear plastic deli containers (11.5-cm diameter × 8-cm tall) lined with moist filter paper. Every 1–2 weeks, we provided each larva with one piece of cat food (Grain-Free Real Chicken Recipe Dry Cat Food, Whole Earth Farm™, Merrick Pet Care Inc., Amarillo, TX, USA), which had been softened in DI-water for 1 h (McLean, Buck & Hanson, 1972). After 24 h, we removed cat food and replaced the filter paper. Occasionally there was extensive fungal growth on the cat food, which could be fatal to Pt. versicolor larvae; in these instances, we gently wiped larvae with DI water and a delicate task wipe then transferred them to clean containers.

Early-instar Pt. versicolor and Pn. pyralis cohorts were reared from eggs laid in July 2020. On the evening of 10 July 2020, we collected 3 male and 2 female Pt. versicolor adults and 3 mated Pt. versicolor females. Flying Pn. pyralis males were collected and identified based on their characteristic “J” flash pattern (Lewis, 2016) while female Pn. pyralis were collected from nearby patches of short grass and were identified based on their flash pattern and similar morphology to the Pn. pyralis males (Lewis, 2016). Female Pt. versicolor were collected near Pn. pyralis females and identified based on their green-shifted flash color and morphology (Lewis, 2016). Additional Pn. pyralis males were collected to provision the mated Pt. versicolor females. We collected Pt. versicolor and Pn. pyralis in a residential area (State College, Centre Co, PA; 40°47′03″N, 77°52′25″W) into two separate 16-oz deli container “nurseries” kept at ambient room temperature (20–22 °C); each nursery contained a handful of moist sphagnum moss on top of moist soil (2-in deep; silt loam, collected from certified organic fields at the Russell E. Larson Agricultural Research Center at Rock Springs, PA, USA; 40°42′52″N, 77°56′46″W). Both Pn. pyralis females mated within a few minutes of collection.

Female Pt. versicolor and Pn. pyralis laid eggs within the following 3 days (50+ Pt. versicolor eggs and 100+ Pn. pyralis eggs; we did not attempt more accurate counts to avoid damaging eggs). Under ambient temperatures (20–22 °C), first-instar larvae of both species began to emerge three weeks after eggs were laid (5 August 2020). We kept Pt. versicolor larvae in the nursery chambers for two weeks, and then, after we observed significant cannibalism among larvae, moved them into individual soil-lined 1-oz polypropylene portion containers. As with larvae collected and reared from 2019, developing Pt. versicolor were fed moistened cat food (Grain-Free Real Chicken Recipe Dry Cat Food, Whole Earth Farm™, Merrick Pet Care Inc., Amarillo, TX, USA) in addition to pieces of freeze-killed Lumbricus terrestris (Josh’s Frogs, Owosso, MI, USA). As evidence of the hypothesis that Pn. pyralis larvae are specialist on earthworms, Pn. pyralis larvae did not feed on cat food but did feed gregariously on freeze-killed L. terrestris. Unlike Pt. versicolor, Pn. pyralis failed to thrive in isolation, so they were kept in the nursery chamber until starting the toxicity assay.

Toxicity assay on Late-instar Photuris versicolor

We started the toxicity assay with late-instar Photuris versicolor on 22 June 2020. We used 1-oz polypropylene portion containers containing 8 g of dry soil (same soil source as nursery chambers) for our assay containers. To the soil in each assay container, we added 0.5 mL of the appropriate clothianidin stock solution, allowed the acetone to completely evaporate, then added two mL of DI water to moisten the soil and to achieve clothianidin concentrations of 0 ng g−1, 101 ng g−1 soil, 102 ng g−1 soil, 103 ng g−1 soil , 104 ng g−1 soil. We chose this concentration range (101–104 ng g−1 soil) to encompass the range of neonicotinoid concentrations in soil that have been measured in potential firefly habitats (Knoepp et al., 2012; Schaafsma et al., 2015; Pearsons et al., 2021).

After setting up assay containers, we weighed the late-instar Pt. versicolor and randomly assigned each to a particular clothianidin concentration (ensuring all larvae in each dose-set were sourced from the same location). All late-instar Pt. versicolor were over 12 months old at the start of the assay, and were over 10-mm long and >50 mg (Table 1). Each clothianidin concentration (0, 101 ng g−1 soil, 102 ng g−1 soil, 103 ng g−1 soil , 104 ng g−1 soil) was replicated six times (30 late-instar Pt. versicolor larvae in total). We recorded firefly status at 1, 4, and 24 h, and every day for an additional 99 d. Fireflies were categorized as dead (D), exhibiting a toxic response (T), or apparently healthy (A). A larva was assumed dead if it did not respond to gentle prodding with forceps. If a larva was flipped on its back and/or demonstrating repetitive twitching of its legs or head, it was recorded as exhibiting a toxic response (T). Fireflies were recorded as apparently healthy (A) if they exhibited a usual response to prodding from blunt forceps (Fig. 1A; quickly curled up on its side, often glowing). During the toxicity assay, we fed larvae once a week by carefully transferring individuals out of the assay containers into clean containers lined with moistened filter paper and containing a piece of moistened cat food. After 24 h, we returned fireflies to the assay containers and noted if the cat food had obvious signs of feeding (Fig. 1B). Feeding activity for each week was measured as a simple binary (0 = no obvious signs of feeding, 1 = obvious signs of feeding). At each status check, we noted if a firefly had constructed a protective soil chamber, then carefully dismantled the chamber to check larval status. Larvae often re-built soil chambers by the next day; if a larva built soil chambers on multiple consecutive days (feeding days as an exception), we noted this behavior as a “period of chamber formation.” Assay containers were kept in a dark drawer except when doing daily checks, and we misted containers with DI water as needed to keep the soil from drying out.

Figure 1 Healthy Pt. versicolor. larvae (A) demonstrating a typical “curl and glow” response after being prodded with blunt forceps and (B) feeding on moistened cat food. (C) An intoxicated Pt. versicolor larva on its back, unable to right itself.

Toxicity assay on early-instar Photuris versicolor

The toxicity assay with early-instar Photuris versicolor was similar to the assay with late-instar larvae, except we added half the amount of soil (4 g) and half the volume of clothianidin stock solutions (0.25 mL) to each assay container to achieve the same clothianidin concentrations (0, 101 ng g−1 soil, 102 ng g−1 soil, 103 ng g−1 soil , 104 ng g−1 soil). All early-instar Pt. versicolor were less than 3 months old and weighed between 3 and 15 mg. On 17 Sept 2020, we started trials with early-instar Pt. versicolor (three replicates at each concentration, 15 larvae in total), feeding them cat food once a week and recording their status at 1, 4, and 24 h, and every day for 10 d, then twice a week for an additional 90 d. Unlike for late-instar Pt. versicolor, we fed early-instars by directly placing moistened cat food in the assay containers (we removed the food 24 h later after noting if food had been damaged [1] or not [0]).

Toxicity assay on early-instar Photinus pyralis

As with the early-instar Pt. versicolor assay, the Photinus pyralis assay was run in 1-oz polypropylene portion containers containing 4 g of soil with 0.25 mL doses of clothianidin stock solutions (to achieve 0, 101 ng g−1 soil, 102 ng g−1 soil, 103 ng g−1 soil , 104 ng g−1 soil). All early-instar Pn. pyralis were less than 3 months old and weighed between 0.6 and 2.4 mg. On 17 Sept 2020, we started the assay on early-instar Pn. pyralis, exposing larvae in sets of five (five larvae per container, three replicates at each concentration, 75 larvae in total), recorded their status at 1, 4, and 24 h, and every day for 10 d, then at least twice a week for an additional 20 d. We terminated the Pn. pyralis assay earlier than the Pt. versicolor assays due to an acarid mite infestation, which rapidly increased larval mortality across all doses. During the assay, we fed Pn. pyralis pieces of earthworm (L. terrestris) in the same manner that early-instar Pt. versicolor were fed cat food.

Statistical analysis

We performed all statistical analyses in R (v4.0.4) (R Core Team, 2021). For each firefly cohort, we calculated median toxic concentrations (TC50) and median lethal concentrations (LC50) at 24 h, 7 d, and 30 d of exposure using probit analysis (LC_PROBIT from the “ecotox” package; Robertson et al., 2017; Hlina et al., 2019); for TC50 estimates, we included both sub-lethal and lethal responses, while LC50 estimates were based on mortality alone. To assess long-term survivorship across clothianidin levels, we used the Kaplan–Meier method (“survival” functions SURVDIFF and PAIRWISE_SURVDIFF; Therneau, 2021; Therneau & Grambsch, 2000). To determine how clothianidin exposure affected firefly behavior, we used non-parametric Mann–Whitney U tests (WILCOX.TEST) to compare feeding frequency and soil-chamber construction across clothianidin doses; we made pairwise comparisons using Wilcoxon rank sum tests with continuity corrections (PAIRWISE.WILCOX.TEST). As firefly larvae reduce feeding before pupation (McLean, Buck & Hanson, 1972), we excluded the two feeding events preceding pupation for feeding assessments.

Results

24 h, 7 d, and 30 d TC50 and LC50 estimates

Dose–response curves and estimated TC50 and LC50 indicate that Photuris versicolor and Photinus pyralis were surprisingly tolerant of exposure to clothianidin (Table 2 and Figs. 2–4). Reliable TC50 and LC50 estimates were limited by our small sample sizes and low acute mortality within the tested concentration range. Overall, TC50 values ranged from 500 ng g−1 to 2,000 ng g−1 while LC50 values exceeded our test limit (above 10,000 ng g−1).

Table 2 Estimated median toxic concentrations (TC50) and lethal concentrations (LC50) for Pt. versicolor and Pn. pyralis exposure to clothianidin-contaminated soil.

95% confidence intervals (CI) are based on probit analyses. CIs are not shown where data did not fit a cumulative standard normal distribution. n.r. = no response in tested range.

Species	timeframe		TC50
(ng g−1 soil)	95% CI		LC50 (ng g−1 soil)	95% CI	
Pt. versicolor,	24 h		1882	136–10,000+		>10,000	–	
late-instar,	7 d		648	144–3047		>10,000	–	
6 larvae / dose	30 d		574	46–9895		>10,000	–	
Pt. versicolor,	24 h		>10,000	–		n.r.	–	
early-instar,	7 d		1169	–		>10,000	–	
3 larvae / dose	30 d		1169	–		1169	–	
Pn. pyralis,	24 h		1726	836–3486		n.r.	–	
early-instar,	7 d		704	–		n.r.	–	
3 sets of 5 / dose	30 d		316	–		1591	246–10,000+	

Figure 2 Dose–response curves for late-instar Pt. versicolor exposed to clothianidin-contaminated soil at 10, 100, 1,000, and 10,000 ng clothianidin per gram of soil (n = 6 larvae for each concentration).

Toxic responses after (A) 24 h, (B) 7 d, and (C) 30 d, and lethal response after (D) 24 h, (E) 7 d, and (F) 30 d. Black dots in each panel represent mean responses at each insecticide concentration; the shaded area represents the 95% confidence interval for each curve. Blue diamonds represent the response of the control group. Dotted lines in each panel marks the 50% toxic response or mortality threshold.

Figure 3 Dose–response curves for early-instar Pt. versicolor exposed to clothianidin-contaminated soil at 10, 100, 1,000, and 10,000 ng clothianidin per gram of soil (n = 3 larvae for each concentration).

Toxic responses after (A) 24 h, (B) 7 d, and (C) 30 d, and lethal response after (D) 24 h, (E) 7 d, and (F) 30 d. Black dots in each panel represent mean responses at each insecticide concentration; the shaded area represents the 95% confidence interval for each curve. Blue diamonds represent the response of the control group. Dotted lines in each panel marks the 50% toxic response or mortality threshold.

Figure 4 Dose–response curves for early-instar Pn. pyralis exposed to clothianidin-contaminated soil at 10, 100, 1,000, and 10,000 ng clothianidin per gram of soil (n = 3 sets of 5 larvae for each concentration).

Toxic responses after (A) 24 h, (B) 7 d, and (C) 30 d, and lethal response after (D) 24 h, (E) 7 d, and (F) 30 d. Black dots in each panel represent mean responses at each insecticide concentration; the shaded area represents the 95% confidence interval for each curve. Blue diamonds represent the response of the control group. Dotted lines in each panel marks the 50% toxic response or mortality threshold.

Firefly survival

Clothianidin exposure significantly reduced long-term firefly survival at high concentrations (Fig. 5). Between one and four hours after initial exposure, half of the late-instar Pt. versicolor larvae and 87% of the early-instar Pn. Pyralis larvae exposed to the highest clothianidin concentration (10,000 ng g−1) began to exhibit toxic responses. By 24 h, all six late-instar Pt. versicolor exposed to the highest clothianidin concentration (10,000 ng g−1) exhibited a toxic response (Fig. 2A); these larvae never recovered and died by day 84. Photuris larvae were somewhat tolerant to lower clothianidin concentrations (10 ng g−1 or 100 ng g−1) and neither late- nor early-instar larvae exposed to low concentrations had significantly lower 100 d survival probability compared to controls (Figs. 5A–5B). All Pt. versicolor in the control treatment either pupated (2 out of 6 late-instar larvae) or survived through day 100 (4 out of 6 late-instar larvae, all three early-instar larvae). Although the experiment was terminated at 30 d due to the mite infestation, early-instar Pn. Pyralis exposed to 1,000 ng g−1 clothianidin showed marginally non-significant reduced survivorship (P = 0.07) while Pn. pyralis exposed to 10,000 ng g−1 clothianidin showed significantly reduced survivorship (P < 0.0001) compared to controls (Fig. 5C).

Figure 5 Survivorship curves.

(A) late-instar Pt. versicolor (n = 6 per concentration), (B) early-instar Pt. versicolor (n = 3 per concentration), and (C) early-instar Pn. pyralis (n = 15 per concentration) at different clothianidin concentrations. P-values next to each line indicate the significance of reduced survivorship compared to the control (with a Benjamini–Hochberg correction for multiple comparisons). P-values were excluded where survivorship was 100% and perfectly overlapped with control values (100 ng g−1 in panel B, 10 and 100 ng g−1 in (C). Survival was significantly affected by clothianidin exposure (late-instar Pt. versicolor: χ42=18, P = 0.001; early-instar Pt. versicolor: χ42=12.5, P = 0.01; early-instar Pn. pyralis: χ42=58.3, P < 0.0001).

Feeding behavior

Clothianidin exposure significantly reduced the number of times firefly larvae fed (Fig. 6). During the toxicity assays, no Pn. pyralis or Pt. versicolor larvae exposed to the highest clothianidin concentration (10,000 ng g−1 soil) fed. Late-instar Pt. versicolor exposed to 1,000 ng g−1 soil fed significantly less frequently than control larvae (χ24 = 16.3, P = 0.003), and early-instar Pn. pyralis larvae fed significantly less at higher doses (1,000 ng g−1 and 10,000 ng g−1) compared to the control or lower doses (χ21 = 12.4, P = 0.0004).

Figure 6 Percent of feeding opportunities taken by firefly larvae.

(A) Late-instar Pt. versicolor larvae (χ42=15.8, P = 0.003), (B) early-instar Pt. versicolor larvae (χ42=8.2, P = 0.08), and (C) early-instar Pn. pyralis larvae (χ12=12.4, P = 0.0004). Different letters indicate significant differences in feeding activity within each cohort at P < 0.05 (Benjamini–Hochberg correction for multiple comparisons).

Soil-chambers, molting, and pupation of late-instar Photuris versicolor

The 14 late-instar Photuris larvae that survived as larvae through day 100 went through 1 to 5 periods of consecutive days when they regularly formed protective soil chambers (median = 2 periods) and spent anywhere from 1 to 20 total days in soil chambers (median = 9 d). Larvae exposed to 10,000 ng g−1 clothianidin never constructed soil chambers while larvae exposed to 1 ppm clothianidin spent significantly fewer days in soil chambers than larvae exposed to 10 ng g−1 (P = 0.01; Fig. 7).

Figure 7 Amount of time that late-instar Pt. versicolor spent in soil chambers at different clothianidin-exposure levels (χ42=18.4, P = 0.001).

Different letters indicate significant differences at P < 0.05 (Benjamini–Hochberg correction for multiple comparisons).

Formation of protective soil chambers did not correspond with molting or pupation, and all recorded molting and pupation events occurred outside soil chambers, on the soil surface. Late-instar Pt. versicolor larvae only molted once or twice, irrespective of how frequently or for how long they built soil chambers (larvae that survived through 100 days; frequency: R2adj = −0.09, F1,10 = 0.10, P = 0.76; duration: R2adj = −0.02, F1,10 = 0.81, P = 0.39). Six of the thirty late-instar Pt. versicolor larvae pupated; five of which successfully eclosed within 35 d of starting the assay (two controls, one at 10 ng g−1, two at 100 ng g−1) and one which was unsuccessful (1,000 ng g−1). The unsuccessful larva failed to shed its last-instar exoskeleton and died during the pupal stage. At 35 d, three of the larvae exposed to the highest clothianidin concentration (10,000 ng g−1) were still alive, but none of these larvae ever entered a pupal stage. Of the individuals that successfully eclosed, three were lab-reared from eggs laid in 2019 (3 out of 5) while only two were wild-collected (2 out of 25).

Discussion

Photuris versicolor complex and Photinus pyralis larvae did not significantly respond to clothianidin concentrations at or below 100 ng g−1 soil, but both firefly species exhibited significant toxic responses to higher concentrations. Although some of the larvae exposed to 10,000 ng clothianidin g−1 soil showed a toxic response within four hours of exposure, compared to other soil invertebrates, larvae of these two firefly species were relatively tolerant to clothianidin-treated soil. Twenty-four hour TC50 values for Pt. versicolor and Pn. pyralis were over 2 × and 30 × the TC50values for the earthworm Eisenia andrei and the collembolan Folsomia candida, respectively (de Limae Silva et al., 2020). Twenty-four hour LC50 values exceeded our maximum test concentration of 104 ng clothianidin g−1 soil (10 ppm), indicating higher tolerance to clothianidin compared to other soil-dwelling beetles (Agriotes spp. [Elateridae] and Atheta coriaria [Staphylinidae]; (Van Herk et al., 2007; Cloyd et al., 2009)). The one other study which tested neonicotinoid toxicity to fireflies observed 13% survival of aquatic Luciola lateralis larvae after 24 h of exposure to 105 ng thiamethoxam mL−1 in water (Lee et al., 2008); these results suggest that fireflies as a group may be somewhat tolerant to neonicotinoid exposure, although this is likely a tenuous conclusion because it is based on just two studies that represent less than 0.2% of all described firefly species (Lewis et al., 2020). Tolerance to neonicotinoids may partly explain why populations of Pt. versicolor and Pn. pyralis do not appear to be declining as fast as rarer firefly species (Reed et al., 2020), which may be more sensitive to neonicotinoid exposure. Pt. versicolor and Pn. pyralis may tolerate clothianidin exposure due to multiple behavioral, morphological, and biochemical processes that could limit their sensitivity to clothianidin (Alyokhin et al., 2008).

Behavioral avoidance of neonicotinoids has been observed across insect orders and beetle families (Easton & Goulson, 2013; Fernandes et al., 2016; Pisa et al., 2021; Korenko et al., 2019), and the results of this current study provide some support for behavioral avoidance of neonicotinoids by Lampyridae. Although firefly larvae could not avoid dermal exposure to the treated soil in our arenas, they may have decreased oral exposure by limiting construction of their soil chambers. To form soil chambers, Pt. versicolor larvae manipulate soil with their mouthparts (Buschman, 1984), providing a potentially more toxic pathway for neonicotinoid exposure (Decourtye & Devillers, 2010). Because neonicotinoids are repellant to other beetle species (Easton & Goulson, 2013), neonicotinoid-treated soil could have repulsed firefly larvae, possibly explaining reduced chamber formation above 1,000 ng clothianidin g−1 soil. Alternatively, sub-lethal neonicotinoid exposure may simply decrease the ability of fireflies to construct soil chambers. Choice-based avoidance studies could be used to test if avoidance or direct toxicity drove the decreased time late-instar Pt. versicolor spent constructing and inhabiting soil chambers at high-clothianidin concentrations.

In addition to behavioral avoidance, specific morphological and metabolic characteristics of fireflies may protect Pt. versicolor and Pn. pyralis larvae from toxic clothianidin exposure. Unlike many other soil invertebrates (e.g., earthworms and mollusks), firefly larvae have a comparably protective cuticle that may act as an effective barrier against neonicotinoid uptake (Decourtye & Devillers, 2010; Wang et al., 2012). And even when clothianidin is absorbed, insects can resist target-site exposure by quickly detoxify and/or excrete neonicotinoids (Olson, Dively & Nelson, 2000; Alyokhin et al., 2008). Although there has been no work on neonicotinoid metabolism by fireflies, Pt. versicolor and Pn. pyralis may upregulate detoxification enzymes after clothianidin exposure, similar to an aquatic firefly species after exposure to benzo[a]pyrene (Zhang et al., 2021). Additionally, Pt. versicolor and Pn. pyralis may be tolerant to clothianidin if neonicotinoids have a low binding affinity to target sites on firefly neurons. Neonicotinoids primarily target nicotinic acetylcholine receptors (nAChRs), which regulate cation movement and neuron firing in response to acetylcholine levels (Matsuda, Ihara & Sattelle, 2020). Neonicotinoid insecticides agonistically bind to these receptors, forcing ion channels open, leading to spasms and eventual paralysis (Simon-Delso et al., 2015). As neonicotinoids have broad activity across insect orders (Matsuda, Ihara & Sattelle, 2020), it is unlikely that clothianidin has a low binding affinity for nAChRs of Pt. versicolor and Pn. pyralis.

There is also the unlikely possibility that extensive neonicotinoid use has exerted selection pressure on the firefly populations in central Pennsylvania to evolve resistance to clothianidin. The way neonicotinoids are currently used is a perfect storm for developing insecticide resistance (Tooker, Douglas & Krupke, 2017), and while most concern has focused on resistance-development in herbivorous pest species, biocontrol agents and other predatory arthropods can develop insecticide tolerance and resistance in response to heavy insecticide use (Bielza, 2016; Mota-Sanchez & Wise, 2021). Although insecticide-resistance is thought to be rare among biocontrol agents, lady beetles (Coleoptera: Coccinellidae) in particular, have been found to develop resistance to a variety of broad-spectrum insecticides, including neonicotinoids (Tang et al., 2015). Insecticide resistance has not been studied in many non-pest species (including lampyrids), but if the selection pressure is high enough, firefly populations could evolve increased tolerance or even resistance to neonicotinoid insecticides.

Differences among any of these potential mechanisms are likely driving differences in tolerance between the two firefly species, namely, the dramatically reduced feeding response of Pn. pyralis to clothianidin exposure. Although this difference could have been exacerbated by mite pressure and the smaller body size of early-instar Pn. pyralis, it is possible that Pn. pyralis has higher uptake, higher active-site affinity, or lower metabolism of clothianidin as compared to Pt. versicolor.

Despite their relative tolerance to clothianidin exposure, field-realistic neonicotinoid concentrations may still pose a chronic threat to Pt. versicolor and Pn. pyralis . Although residual neonicotinoid concentrations in soil are often below 100 ng g−1 (Schaafsma et al., 2016; Radolinski et al., 2019; Pearsons et al., 2021), concentrations can regularly exceed these levels after agricultural applications (as high as 594 ng g−1 23 days after planting neonicotinoid-coated seeds; (Radolinski et al., 2019)), after turf applications (3 × higher than in agronomic settings; (Armbrust & Peeler, 2002)) and after soil drenches to manage hemlock wooly adelgid (over 4,000 ng AI g−1 soil; Knoepp et al., 2012). Such high concentrations are well within the acutely toxic and chronically lethal range for Pt. versicolor and Pn. pyralis larvae (Table 2). Encountering such high concentrations are likely to be even more lethal under field conditions, as firefly larvae that exhibited toxic responses in the laboratory would be vulnerable to predation and starvation, two risks that can increase mortality from insecticides (Kunkel, Held & Potter, 2001). Additionally, further work is needed to assess if neonicotinoid exposure can exacerbate other stressors affecting firefly populations (i.e., light pollution) or if neonicotinoids pose a significant risk to firefly eggs or adults.

As observed with other predatory beetle species (Cycloneda sanguinea [Coccinellidae] and Chauliognathus flavipes [Cantharidae]; Fernandes et al., 2016, firefly larvae exhibited reduced feeding activity in response to high neonicotinoid exposure. Firefly larvae that feed less frequently may have less successful eclosion rates, and those that do eclose may have lower reproductive success. Additionally, the prey that fireflies encounter in neonicotinoid-contaminated environments likely provide an additional neonicotinoid exposure route. Photinus larvae primarily feed on earthworms (Lewis et al., 2020), which have been found to contain neonicotinoid concentrations above 200 ng g−1 when collected from soybean fields that were planted with neonicotinoid-coated seeds (Douglas, Rohr & Tooker, 2015) and 700 ng g−1 when collected from treated cereal fields (Pelosi et al., 2021). Firefly larvae of other species are known to feed on slugs (Barker, 2004), which can also contain high doses of neonicotinoids (500 ng g−1), leading to disrupted biological control provided by carabid beetles (Douglas, Rohr & Tooker, 2015). Compounded with reduced prey availability in habitats where neonicotinoids are used (Ritchie et al., 2019; Tooker & Pearsons, 2021), decreased feeding activity and high risks of further neonicotinoid exposure through contaminated prey may explain why adult lampyrid densities are significantly lower where clothianidin has been used as a seed coating (Disque et al., 2019), even if acute mortality is low. Adult fireflies may also encounter neonicotinoid residues while resting on sprayed vegetation or during oviposition into soil (Pisa et al., 2021), although the risk of such exposure does not appear to have been explored.

Despite low acute mortality, the sublethal effects of clothianidin were surprising, as some Pt. versicolor larvae survived in a severely intoxicated state (not feeding, not building protective soil chambers, only occasionally moving legs and/or mandibles) for over two months. A similar phenomenon has been observed in European wireworms (Agriotes spp. [Coleoptera: Elateridae]) after exposure to clothianidin, with individuals surviving and even recovering from a severely intoxicated state that can last months (Van Herk et al., 2007; Vernon et al., 2007). For pests like Agriotes spp., such sub-lethal effects of clothianidin exposure could still decrease crop damage but may exacerbate the risk of Agriotes spp. developing neonicotinoid resistance. For predators like Pt. versicolor, this long-term intoxication may limit their potential to provide biological control beyond what would be expected based on population declines.

Conclusions

As larvae of the two firefly species that we studied appear to be somewhat tolerant to clothianidin-treated soil, neonicotinoids alone may not be significant direct factors in firefly declines in North America, at least for common species. Nevertheless, firefly populations around the world appear to be suffering from other stressors (e.g., habitat loss, reduced prey availability, light pollution), and ecological research has demonstrated that animal populations exposed to multiple stressors can suffer disproportionally more than what is suffered from a single stress (Relyea & Mills, 2001). Therefore, continued widespread contamination of larval firefly habitats with neonicotinoids may hold potential to exacerbate the influence of other stressors on declining firefly populations (Lewis et al., 2020). We encourage researchers with access to other species of fireflies, particularly those with declining populations in areas where neonicotinoids are commonly used, to explore their toxicological responses to insecticides.

Supplemental Information

Supplemental Information 1 Raw Data

Firefly larvae acute responses, long-term survival, feeding behavior, soil chamber formation, and molting frequency under clothianidin exposure.

Click here for additional data file.

We would like to thank Cheyenne McKinley and Nellie Heitzman for assistance with specimen collection and rearing. We would also like to thank Larry Buschman for his advice on firefly collection and rearing.

Additional Information and Declarations

Competing Interests

Author Contributions

Data Availability

The authors declare there are no competing interests.

Kirsten Ann Pearsons conceived and designed the experiments, performed the experiments, analyzed the data, prepared figures and/or tables, authored or reviewed drafts of the paper, and approved the final draft.

Sarah E. Lower and John F. Tooker conceived and designed the experiments, authored or reviewed drafts of the paper, and approved the final draft.

The following information was supplied regarding data availability:

The raw data is available in the Supplementary Files.

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
