# Peer review of "Toxicity of clothianidin to common Eastern North American fireflies"

_PeerJ, doi:10.7717/peerj.12495_

## Round 0.1 · original submission · Minor Revisions

All three reviewers have concluded that your manuscript only requires minor revisions. The reviewer comments are clear, so I won't repeat or emphasize them here. Please address the reviewer comments and modify your manuscript accordingly.

·

Basic reporting

no comment

Experimental design

The number of samples are so limited; however, the observation took long time and can conclude the low risk of the chemical. It is the novel knowledge.

Validity of the findings

no comment

Additional comments

Abstract – no comment

Introduction – no comment

Materials & methods

- You gave the good information of firefly collection (lines 78-87). If you arrange them in a table, it might help the reader understand your firefly collection easily including the numbers of larvae of 2 testing groups.
- Please define more detail of 2 groups of testing larvae. Which instars was classified for late-instar larvae and early-instar larvae? Are they classified by moulting or size? If it is size, please define size range.
- I think you can use genus abbreviation for the second writing. As in https://doi.org/10.3390/genes11060627, they used the different Pt. and Pn. You can check from other references.
- On lines 90 “…while the majority of late-instar Photuris were wild-collected in summer of 2019”, Could you tell the numbers of them?
- On line 180, please explain more. How to record feeding frequency for the analysis.

Results

- From the methods, the responses were observed at 1 and 4 h after exposure. Could you explain more when the larvae began the responses on line 195?
- In the part of “Feeding behavior” on line 204. There was pupation of tested larvae in some cases of late-instar larvae. This might affect on feeding behavior because the larvae before pupation will not show feeding behavior. Did it affect on any results in this part?
- On line 213 (1 to 5 peroinds), do the authors mean "larval instar"? If it is, please use the words for more appropriate meaning.
- On lines 214-215, please put “unit” for the median.
- Do you have the information of abnormal moulting behavior?

Discussion

- Your discussion needs more detail about mode of action of clothianidin. You can improve your discussion on the toxic responses.
- It is the good idea to discuss the behavioral avoidance (construction of soil chambers) to the chemical on lines 245-247. I wonder if the larvae were in unsuitable environments such as too dry soil. Did you control the soil moisture? If the larvae live under dry substrate, do they construct and live in soil chambers? If you controlled it, you can discuss their avoidance.

Figures
Fig 2-4. The data of control groups might be needed for the analysis. They should be in the graphs. Could you improve them?

·

Basic reporting

This paper addresses a quite important question concerning how neonicotinoid insecticides might impact the terrestrial larval stage of North American fireflies. The manuscript is well-written, provides relevant background, and clearly articulates the research question; in addition, data are clearly presented in six figures and one table.

I’ve made several suggestions on the pdf for points that I think should be clarified (sorry, pdf issues impeded comments after p14), and I’ve summarized these minor suggestions below in hopes that these might help improve the manuscript.

Experimental design

Minor comments:

1) Some additional information for methods:
a) It would be useful to know how you chose the specific clothianidin concentrations used in these experiments (lines 136-7), and how these relate to levels likely to be encountered in the field. This important context is now presented in the Discussion (lines 288-294), but curious readers might like to hear about this in the Methods.
b) It would be helpful to know the size range of the larvae that were used in each experiment (lines 134, 156, 165).
c) While sample sizes are given here, it might be helpful to reiterate these in the results (noted in pdf).

Validity of the findings

Minor comments:
2) Results
a) More consistency is needed in reporting treatment levels (=clothianidin concentrations): in places you switch from reporting treatment levels in ng per g soil to ug per g soil (eg. line 200), and then also switch sometimes to ppm or ppb (eg. lines 189). Not sure of the rationale for this but it seems unnecessarily confusing – clarity of the MS would benefit greatly from using consistent measurement units that match the data shown in your figures (eg. ng g-1)
b) In lines 200-2, you should remind readers that mites prematurely ended this one experiment.
c) Figure 5 (survivorship curves): where sample sizes are given, add “per treatment”

Additional comments

Minor comments:

3) Discussion
a) In the first paragraph it would be worthwhile to compare the toxicity results to those seen in previous studies of the aquatic larvae of Asian fireflies (cited in the Intro).
b) Here again, please convert units to those used in figures.
c) In the discussion it might be worthwhile to mention possible adult exposure for adults resting on vegetation and for females ovipositing in soil.

4) Very minor comments:

a) 3) Figure 1A currently shows the startle response of a healthy Photuris larva. If available, it might be informative to include a contrasting image of a larvae exhibiting a toxic (T) response.

b) 4) In the review of previous work on effects of insecticides on the aquatic larvae of Asian fireflies (lines 47), please include species names.

c) 5) Any information on sex ratio of the Photuris that successfully eclosed?

Reviewer 3 ·

Basic reporting

This is an important topic that was well-prepared by the authors. It’s funny, I’d been wondering about fireflies and neonicotinoids for a number of years now; glad someone got around to evaluating this group. Since they don’t naturally fall into the ecosystem service category like pollinators or natural enemies, fireflies have fallen to the wayside a bit, but people clearly care about them from an aesthetic perspective.
The writing is very good throughout and I only have some minor wording suggestions noted. The authors cite appropriate studies to support their findings, background, and rationale. I also had no problem interpreting the figures.

Experimental design

The methodology was appropriate to the question. I liked the approach of dosing up the soil to test the larval stage. The authors did a nice job acquiring firefly offspring and rearing them in the laboratory, which is not all that easy to do. However, would’ve been better to measure clothianidin in the soil to confirm the target residues.

Validity of the findings

There’s no doubt that the sample size is very low and this study’s conclusions would be stronger if they had higher replication. Also, the authors test the larvae over a 30-100 day period, but in nature these insects are belowground for 1-2 years, so they’re only capturing a snapshot of the lifecycle, which is fine for short-term toxicity testing, but sub-lethal effects are harder to tease apart, assuming that these negative effects would accumulate over time spent in contact with the chemical.
That being said, I’m more lenient on this issue due to the fact that these data are hard to come by and it still seems likely based on their findings that fireflies are relatively tolerant of clothianidin compared to other insect groups.

Additional comments

General Comments…
Just as a general comment, the authors found weak effects of clothianidin on fireflies at field-relevant concentrations, yet still try to concoct a path by which neonicotinoids are involved in their decline in the Discussion. I think this is a general problem with eco-toxicology literature, i.e., authors try to find a way to conclude that a certain pesticide is involved in species declines, even if their data do not support this (in which case, why conduct the study to begin with?). I understand that they can’t exclude interactive effects with other stressors, but that’s not what they tested and that’s always a caveat thrown at the end of eco-tox studies as a means to justify weak responses. I realize the authors are attempting to be cautious and don’t want to flat-out say that neonics are perfectly safe for fireflies, but the alternative is also problematic and a bit of a pet-peeve of mine for this type of literature. Just something to be careful of in the writing.

Line Edits…
L14 & 19. It’s unclear at first-read (I understand now after reading the whole paper) why you’re specifying the larval stage here. I’d either be more specific about why larvae would be more vulnerable than other life stages, or use more general language about firefly exposure.
L21. I’d remove one of the two uses of “larvae” (probably the first) to avoid redundancy.
L21-23. Would be good to also mention what your response variables were.
L24. I’d avoid making comparisons with other insect groups, at least in the Results section. Your Methods don’t say anything about testing other non-firefly insect groups so this is interpretation that would be more appropriate in the Discussion sub-section beneath.
L24 & 25. Genus name should be abbreviated on second use. Here and throughout the rest of the paper as well. Unless that’s against PeerJ policy.
L29-31. Mmmm, I’m having a hard time squaring this conclusion with the statement immediately prior (L24-25). First they’re relatively tolerant and then it could contribute to population declines? Seems like mixed-messaging bit. And related to my broader comment above.
L82. Some of this info could be useful in the Discussion for interpreting why the authors found such high tolerance. Larger species tend to be more tolerant of insecticides. Also, are these species of fireflies currently in decline? This would be helpful to know. It says they’re a widespread species, but unclear if they’re undergoing a range-contraction. If not, this could also explain the tolerance (i.e., rare/declining taxa could be more vulnerable).
L88. Can you be any more specific about what late-instar means (i.e., how you determine larval instars from field-collected individuals)? How many larval instars do they even have?
L90. Does this mean you wild-collected the larval stage? How? By sifting through soil? Can you even ID the larvae? More info here would be helpful.
L96. Is there a citation for the use of cat food in firefly rearing?
L131. I assume the soil was then mixed around? Or did it just go on the soil surface?
L136-137. Why these concentrations? This could use more rationale. A “normal” range of soil residues were reported in the Intro (L55), but these were in different units and this section could still benefit from explaining why you chose these specific doses.
L137, 155, 164. You’re using the term “assay” I think in a way that I’m unaccustomed to. I would refer to an assay as synonymous to “trial”. But it looks like you’re using it here more to mean a block or replicate? Please try to clarify.
L230 (Discussion). I realize you only studied the larval stage, but it might be helpful to briefly note possible routes of exposure for the adults as well.
L316. You just said “despite low acute mortality” on the prior line; looks a little strange back-to-back
L399. Misspelling of “Journal”

---

## Round 0.2 · accepted · Accept

Thank you for revising your manuscript following reviewer comments and your thoughtful responses to those comments.